# Leadership Opportunities in the School Setting: A Scoping Study on Staff Perceptions

Robert Hannan, Niamh Lafferty *  and Patricia Mannix McNamara

School of Education, University of Limerick, V94 PX58 Limerick, Ireland
* Correspondence: Niamh.Lafferty@ul.ie

**Abstract:** The focus of this study was to explore teachers' and middle school leaders' perspectives of promotional policies and practices within the schools where they work. As this was an initial scoping study, a qualitative approach was adopted. Fifteen teachers and/or middle school leaders participated in semi-structured interviews. Thematic analysis was employed for data analysis. Themes that emerged from the data included a mix of perceptions, in that promotions were sometimes perceived to be based on appropriate measures of merit such as experience, but at other times were perceived to be unfair or based on cronyism, with female staff perceived to be at a disadvantage. Reasons for seeking out promotion were identified as predominantly being for personal ambition and increased salary. A dark side of promotions also emerged, and this referred to the breakdown of relationships with co-workers following promotions and implications for turnover. Implications for practice, policy, and research are discussed.

**Keywords:** educational leadership; culture; promotions; career advancement; middle leadership





## 1. Introduction

Promotions can be viewed as a product of organisational culture as they act as an indicator of organisational values [1]. Employees may look towards those who have been promoted with questions about what skills or qualities the successful candidate possesses to be deserving of promotion [1–3]. Promotions are integral to the very fabric of an organisation. They are indeed common practice but are often limited due to the pyramidal structure of organisations [4]. As such, there is only room for a select number of people to be promoted within an organisation, regardless of how many may be perceived to deserve one [4]. This is particularly true in the school setting, where there is less employee mobility. In light of this, opportunities for promotion are limited and can then become fraught. Whether accurate or otherwise, where there are limited opportunities for promotion and where competition is intense, the factors that determined who was successful may be perceived as meritless. Organisational leaders are also under pressure to have a "goodness of fit" in those who are promoted. Some may use promotion practices as a means to reward those who align with organisational values and goals [1] and/or to retain staff members [5,6]. Promotions are a complex organisational practice that has received relatively little attention in the research pertaining to the school setting.

When perceived to be fair, the promotion process can act as a motivational tool for employees to incentivise them to work harder and more efficiently [7,8]. Promotions also have the potential to promote job satisfaction [9]. However, when promotions are perceived to be unfair, the resulting sense of procedural injustice may have consequences for discouraging work efforts [10,11], causing job stress [12], and influencing turnover decisions [12]. In the context of school leadership, this is particularly relevant as a staffing crisis in school principalship has been identified, with fewer individuals seeking promotion and increasing school leader turnover [13]. It is, therefore, imperative to explore the perceptions of teaching staff on the motivators, barriers, and facilitators of workplace promotions.

### 1.1. School Leadership and Promotions

The attrition of principals is an emerging phenomenon in school leadership. In the United States, for example, a study conducted in 2015 indicated that among all public-school principals studied, six percent moved to a different school and ten percent left the principalship [14]. In 2021, the numbers remain similar or even slightly increased in terms of intention to leave school principalship, with 13% strongly agreeing and 15% somewhat agreeing that they plan to leave the principalship as soon as they can [15]. In Ireland, a recent national study published by the Irish Primary Principals Network [16] found that in the last five years, there has been a change of leadership in 39% of the smaller schools who responded to their study and in 40% of the DEIS schools that responded. Forty-five percent of principals responded that it was unlikely or highly unlikely that they would be in their current role on five years, in addition to some giving reasons such as retirement, others cited hoping to be seconded to another role, to move schools or that they were not thriving in their current schools and intended to resign [16]. This indicates that in coming years, the growing trends of promotion opportunity and career mobility will continue to grow, and as such it is essential to bolster confidence in recruitment and promotional processes in order for sustainable leadership to thrive.

### 1.2. Reasons for Seeking Promotions

There are numerous reasons why employees seek out career progression. Firstly, intrinsic motivations [17] such as intellectual stimulation [18], a desire to learn more [19], and the opportunity to explore new roles [18,20] may act as motivators. The motivations may also lie in the hierarchical assumptions of seniority such as desire for greater organisational and/or occupational power [21,22] or a desire to lead [23]. Additional reasons may be more extrinsic in nature, such as greater job security [24], pay increases [25], and prestige [25]. These numerous motivations for career progression have been shown to differ across individuals based on gender [21,22,26], nationality [26], culture [27,28], and even simply differences in promotional aspirations across individuals [29]. In considering the numerous reasons as to why individuals seek out promotions, it is important to acknowledge that there is little literature in the existing literature pertaining specifically to motivations of school staff, a gap which is even larger when considering an Irish perspective. The current research seeks to bridge these gaps to identify why staff in this population seek out promotions. This is particularly important as a current staffing crisis has been identified in the role of school leadership [30,31], and to identify motivators could allow for these factors to be considered and embedded in progression opportunities.

### 1.3. Determinants of Promotion

Ruderman et al. [32] have identified that the reasons for promotion success can typically be broken down into five categories: "preparation, attitudes, people skills, personal attributes and context". Internal change agents are typically managers or employees who have been deployed to oversee change within their organisation [33]. In many innovative organisations, leaders and employees are expected to lead organisational change [34]. Frequently what is desired in candidates is a proven ability to be an effective change agent. [32]. Much time within organisations is work relating to others and that interpersonal dimension is frequently underestimated [35]. Successful leaders have a more natural or highly developed ability to read the behaviours of others [36]. This is a critical skill for middle leaders as, quite often, they are expected to manage up as well as manage down [37]. These interpersonal skills and ability to work in a team, which have in effect come to be known as social competence [38], are often viewed by promoting leaders as crucial to success [39]. In pondering why some people experience more career growth, Chen et al. [40] suggest that employees who have congruent characteristics with their organisations are more likely to be rewarded within the organisation by means of a promotion. In this way, promotions can often bolster and protect current organisational culture.

*1.4. Work Politics*

Workplaces are social places [41]. Leaders and followers often discuss non-work-related topics that can sometimes extend outside the working hours over lunch, evening socialising, or smoking breaks. Through these interactions, social bonds occur [41]. Networking behaviours, such as going out to talk shop informally, attending conferences, or keeping in touch with former colleagues, are essential to workplace mobility [42–44]. In line with this, the social nature of promotions has previously been acknowledged, leading to the identification of "work politics" as an indicator of promotional decisions within an organisation [45]. A political promotion is based around social negotiation between leaders and their followers [46,47] and heavily influenced by the interpersonal strand and connections. In a study of promotions, a person's proximity to the hiring manager and/or the hiring manager's superior was evident in 73% of promotions given [48]. Clearly, proximity to and familiarity with the hiring manager has been found to increase the likelihood of promotion, as found by London and Stumpf [49]. These social and political factors can then give rise to actual or perceived unfairness in the promotions processes.

Networking goes hand in hand with accelerated growth in career success as networkers [50]. In a politicised culture, perceived unfairness in promotions can be associated with social bonds and networking with managers and has also been found to have an adverse impact with regard to gender gaps in career development [41]. The gender gap challenges in career promotion are now well-established in the extant literature. Gender quotas on interview boards and unconscious bias training has now become common in many countries when engaging with recruitment. The literature indicates that women need to "push" more for a promotion than their male counterparts [32]. This "pushing" is described as playing "a major role in convincing the boss or company that they deserved a promotion" [32] (p. 8). The concept of "sticky floors" was observed by Booth, Francesconi, and Frank [51] to describe the challenges facing women in gaining promotion and/or career mobility. Obligations at home, lack of time to commute long distances due to home commitments or family, and not able to move for new job opportunities or promotions were reasons cited within the study. Women have a higher turnover rate in the workplace and are less able to signal their skills to their promoting superiors compared with their male counterparts. Caught between these "sticky floors" and "glass ceilings", significantly more women than men remain static in the workplace.

It is widely acknowledged that career advancement in schools, such as moving to principalship, is in crisis, and much of that is attributed to the nature of the work, such as inordinately heavy workloads and poor resourcing. However, the role that promotion processes play and teachers' perceptions of their dynamics is as yet not well understood. Therefore, this scoping study was undertaken to shed light on these factors and to prompt a discourse on promotion practices both perceived and experienced. Specifically, this research aims to identify factors that are associated with the motivations and perceived barriers for seeking promotions in schools. By identifying these factors, future policies can be adopted to design progression opportunities in line with common motivators for promotion and limit any negative perceptions which may prevent staff from seeking out promotion. These are particularly important steps in tackling the staffing crisis in principalship [30,31].

## 2. Materials and Methods

*2.1. Research Design*

This research was qualitative in nature, consisting of semi-structured interviews with 15 participants. These participants were teachers and middle leaders currently working in schools in the UK, Ireland, and Qatar. This research was carried out in alignment with a constructivist approach in which the research teams created knowledge in collaboration with the participants.

## 2.2. Participants

Participant information is highlighted in Table 1. It is important to note that although a number of participants worked in Qatar, none of the participants are citizens of Qatar or raised here. All had moved to Qatar for the specific purpose of work opportunities.

**Table 1.** Participant characteristics.

| Participant | Gender | Position | Studied in | School Type | Works in |
|---|---|---|---|---|---|
| Anna | F | Middle Leader | UK | Private British International | Qatar |
| Michelle | F | Middle Leader | UK | Private British International | Qatar |
| Patrick | M | Middle Leader | Ireland | Private British International | Qatar |
| Rose | F | Middle Leader | USA | Private American International | Qatar |
| Anthony | M | Teacher | UK | Academy | UK |
| Sarah | F | Teacher | Ireland | Private British International | Qatar |
| Barry | M | Teacher | Ireland | Community College | Ireland |
| Matthew | M | Teacher | UK | Free School | UK |
| Aoibheann | F | Teacher | UK | Private British International | Qatar |
| Megan | F | Middle Leader | UK | Private British International | Qatar |
| Emma | F | Middle Leader | UK | Faith School | UK |
| Marie | F | Middle Leader | Ireland | Community College | Ireland |
| Elliot | M | Teacher | UK | Private British International | Qatar |
| Allister | M | Middle Leader | Ireland | Academy | UK |
| Kate | F | Teacher | UK | Private British International | Qatar |

## 2.3. Participant Recruitment

Recruitment took place via an email to principals of several different schools across a range of countries, which would then be passed onto staff members within the school who were not at a senior leadership level. The schools were identified by means of snowball sampling. Close contacts of the research team who are known to work with school leaders were asked to forward an information letter to them via email, which included the contact details of the research team. This added an additional level of confidentiality and to minimise feelings of coercion which may potentially be experienced by the school leaders. This approach resulted in the research team not knowing exactly who had been asked to participate (and who had denied participation) and making anonymity and confidentiality in participation and non-participation greatly enhanced. If contacted by the principal with an agreement to participate, staff participants were gathered through the use of a recruitment letter which outlined the aims of the study and the approach which would be used. This recruitment letter was distributed by the school principal. Within the letter, the researcher team outlined that a semi-structured interview would take place in order to gather information necessary for the study. The recruitment letter was also accompanied by a participant information sheet which outlined the aims of the study, the approach that would be used to gather information, the benefits of the study, the risks of the study, the voluntary nature of participation, and how the data would be stored. The participants were also given an ethical consent form, should they choose to participate in this study, which required a signature of intent for their participation. This ethical consent form again outlined the nature of the respondent's participation in this study.

## 2.4. Instrument Development

To ensure reliability and validity of findings, the development of an interview schedule in line with specified steps has been identified as vital for rigorous research [52,53]. Due to this, the research team followed steps outlined by Kallio et al. [53] for instrument development. These are as follows:

1. Identifying the prerequisites to use a semi-structured interview
   a. As gathering the perceptions and experiences of participants was the focus of the current research, and a constructivist position formed the foundational

roots, semi-structured interviews were identified as relevant for the current study [53].

2.  Retrieving and utilizing previous knowledge

    a.  This was done through a comprehensive literature review and through discussion across the research team and additional experts in educational leadership research. This allowed for the identification of what is known, what is not known, and how to ask questions in this area.

3.  Formulating the preliminary interview guide

    a.  This was achieved through a process of initial drafting by two researchers and followed up with additional conversation and agreement across all members of the research team.

4.  Pilot testing

    a.  The schedule was piloted with retired teachers who were similar to the desired sample but excluded due to their retired status.

Following these steps, a finalised interview schedule was developed that included the addition of prompts and probes to assist the interviewer. Sample of the questions are as follows:

1.  Why do you think people get promoted in your school?
2.  What factors do you think are the most important for influencing promotions in your school?
3.  Are there any benefits to promotion in your opinion? What are they?
4.  Are there any negative consequences to being promoted in your opinion?

The questions were designed to be relatively broad and open-ended to minimise bias and promote dialogue.

### 2.5. Data Collection

The participants themselves were able to identify a location where they would feel most comfortable in meeting for the interview process. These sites ranged from coffee shops to hotel receptions to school offices. The interviews' average length was 43 min. Interviews covered several broad topics throughout, which included: factors influencing promotions; motivations for promotions; benefits and consequences of promotions; work relationships/dynamics; gender gaps; and mentorship. All interviews were recorded on a Dictaphone. Of the 15 interviews that took place, 12 of these were carried out in a face-to-face format. The remaining three interviews were carried out using Microsoft Teams, a GDPR-compliant platform.

### 2.6. Data Analysis

Using Creswell's [54] method for data analysis, the following steps were carried out:

1.  Organise and prepare the data for analysis.

    a.  Pseudonymized transcripts were created and stored on a GDPR-compliant platform.

2.  Read through all the data.

    a.  All transcripts were read individually, at least two times, to allow for familiarity with the text.

3.  Begin detailed analysis with a coding process.

    a.  Line-by-line coding was carried out. Each sentence of the transcript was read and coded for meaning. This process was carried out on each transcript individually a minimum of two times. The coding process was carried out by two of the authors independently of each other, and then comparison was made and agreement reached through discussion. The third author reviewed a subset of the final codes for the purpose of (dis)agreement and further validation. Confirmation was reached across all researchers.

4.    Use the coding process to generate a description of the setting or people as well as categories or themes for analysis.

     a.    Codes were compared and grouped for the purpose of theme development. Themes were discussed and agreement reached across all authors.

5.    Advance how the description and themes will be represented in the qualitative narrative.

     a.    Through discussion, the presentational structure of the themes and aligning codes were identified to ensure the greatest level of clarity for readers. This was to ensure that participant voice was presented in the most accurate way possible.

6.    Making an interpretation or meaning of the data.

     a.    The themes and aligning codes were interpreted for meaning through discussion with all of the research team.

### 2.7. Validity and Reliability

To promote reliability and validity of findings, the research team adopted the same steps as Lynch et al. [52] for inter-reliability, in which members of the research team independently coded sections of the transcripts and compared findings. Agreement was reached through discussion.

### 2.8. Ethical Concerns

Primary concerns were confidentiality and anonymity in participation. To achieve this, participants' names were pseudonymized in the transcripts and any identifying factors were redacted. In discussing aspects such as promotions, it was also a concern that participants may become upset or angry when discussing past experiences of promotions or failed promotion attempts. This was given great consideration in the planning and conduct of interviews, with interview questions designed to minimise risk and maximise awareness of participant wellbeing in the interviews. No participant became angry or upset during the interview process. Additionally, as school principals were made aware through the recruitment process that teachers in their schools might have been participating in their research, it was of high importance that the schools themselves were not identifiable to the school leader, as this may cause upset to themselves should their staff indicate promotion practices in their school were unfavourable. For this reason, aspects such as school size and specific location were not included in the collected data.

## 3. Results

The data analysis procedure outlined in detail in the methodology section of this paper resulted in three prominent themes: (1) why people seek promotions, (2) why people are promoted, and (3) workplace relationship/dynamics stemming from promotional decisions in schools. These main themes will be discussed in accordance with aligning subthemes and supported with a selection of participant quotes. In many cases, these quotes are not an exhaustive list of all participant references to subthemes and themes, but a sample of them which best reflect the stated findings.

### 3.1. Why People Seek Promotional Opportunities in Schools
3.1.1. Pay

Pay was the most frequently identified and discussed reason for seeking out promotions. Remuneration and increased salary were clearly identified as significant motivating factors as perceived by all the participants. For example, Michelle (Qatar) spoke about the fluidity of schools in the Middle East and how extra money for a promotional position can be attractive to some staff.

> What I've seen they have promised more money . . . but in an international school it's unique isn't it because you're not going to stay forever, it's fluid there's a lot of fluidity, people come and go so I think they dangle carrots at people to get them (Michelle, Qatar)

> Some people could get promoted just for financial gain if that's their main priority (Sarah, Qatar)

> I think anyone would be kind of lying here if they said that financial wasn't a thing (Barry, Ireland)

Apart from perceptions of other people's motivations as were expressed by the participants in the above quotes, personal motivations for promotions were also discussed in line with the desirability of pay increases. Having the influence of more money was appealing to several of the participants within the study. Many identified the fact that they would not be willing to take on such positions if it were not for the financial incentive involved. For example:

> My first head of department job I took on an extra 10,000 pounds per year so that was a factor because I wouldn't have been doing such a big job for a small amount of money, so I think that makes a difference in terms of whether they go for that promotion or not (Rose, Qatar)

> I mean pay, pay is a motivator—people might think right so I'm at this point in my career I've reached maybe the top teaching band I want to push on and get more money. (Anthony, UK)

> Obviously, money, more money (Marie, Ireland)

Barry (Ireland) and Elliot (Qatar) further explained the motivation of money, in relation to the impact more money would have on their own personal circumstances:

> I'm a younger person hoping to start a family and buy a home and all these kinds of expensive things, so I mean I'm constantly thinking financial (Barry, Ireland)

> Probably salary I would think because again if you're getting more money then, you can potentially have a better lifestyle etc (Elliot, Qatar)

Interestingly, while money was identified as a motivational factor for promotions, four participants (Barry, Aoibheann, Megan, and Marie) also expressed frustration at how little the money was for the job that was being done. For example:

> At the moment like another 800 riyals, so for 200 euro [per month] it's not a lot for what for what I do with the extra work and that you do, you know (Marie, Ireland)

> Money. Pay definitely of course if it is with the money to be had then yeah, I think it'll be more likely and if it reflects obviously in the job that we're in which I don't think the pay is ever going to reflect the amount of work that we do [as teachers]. (Kate, Qatar)

Additionally, participants discussed not only a disparity between pay and work, but between work roles, responsibilities, and pay as highlighted in the following quotes:

> So we're teachers, we naturally network and are outgoing and will talk with a lot of other teachers and therefore you get to know that people are doing a similar role to you on a lot more money (Emma, UK)

> When you look at pay the pay in comparison to roles and responsibilities it's not equal, so it's not recognised in that sense (Anna, Qatar)

These findings together indicated that pay was identified as a personal motivator for seeking out promotions as well as a perceived motivator for others, despite the acknowledgement of a work–pay disparity. These sentiments were noted by the following eight participants: Anna, Michelle, Patrick, Rose, Anthony, Barry, Matthew, and Elliot, highlighting shared views between males and females.

### 3.1.2. Intrinsic Motivations

A number of intrinsic motivations were identified as motivators for promotion. Firstly, ambition was identified by Allister (UK) as "the main reason people are promoted" which was supported in the following:

> Those that have ambition and work hard and impressed in their time as an ordinary teacher whenever the opportunity arises to go for that job then the proving themselves so therefore, they get that opportunity to step up (Patrick, Qatar)

> When a position came up and I showed interest in it through the process. An English lead came up at the same time and you know we were put through the rigorous process of selection even though there wasn't that many running for it I think our schools still like followed the same process and wanted to see that ambition. (Allister, UK)

The intrinsic motivations alluded to in this study also included generic references to personal motivations for leadership positions or career advancement as highlighted in the following:

> I think it depends on personal motivation like motivational factors for you as a person where you are at in your career. So, for me personally now I'm at a bit of a crossroads in terms of I've done, a couple of leadership opportunities, done some whole school leadership but I kind of want to go more down and teaching and learning path so that's motivating me (Anthony, UK)

> If there is a role that's coming up or a role that someone has been doing and you think yeah, they've done that alright but I can do that better so I can fill that post better. I can take the school forward so I think there's factors there that are quite different motivators, but all may become from intrinsic reasons (Michelle, Qatar, UK)

The above indicates that promotions can be sought out for intrinsic reasons such as ambition or personal desire to further one's career.

### 3.2. Why People Receive Promotions in Schools

Apart from exploring why people seek out promotions, evidence also emerged in relation to why people are successful in their promotion journeys. Experience, cronyism, gender, and personality/credentials were identified as factors influencing the promotion of teachers.

### 3.2.1. Experience

Participants identified experience as the most important reason as to why people receive promotions and as the reason for a potential candidate getting their "foot through the door" when it comes to getting opportunities for interviews. This was highlighted in the following sample of quotes:

> Experience—it would also be a key factor in sense that if you have something that you have done before that even I suppose if you're certified in it, be it something around logistics something around special education needs something around sports if it was pertinent and that it would be a help (Barry, Ireland)

> they're going to look at your experience and your relevance to that role first (Emma, UK)

Participants further indicated that experience will be considered at various stages in the promotion process, including on one's CV and their eventual progress through the promotion process as highlighted in the following quotes:

> I would say it's experience because if you go for another role in a new job, your CV has got to say why you're capable of this new role (Megan, Qatar)

> I think actual first-hand experience and your, like, your teaching experience is going to get you in a lot, it's going to get you far because it shows what you put into practice. (Sarah, Qatar)

One participant spoke about how at times there may be a desire to give people experience by offering a promotional role, and they also acknowledged that if a candidate did not have any prior experience, it would be difficult to reward an applicant with that position:

> It's like a double-edged sword you want to give people the experience but whether they're experienced enough to do the job and then there's that idea of how they are supported to do that job as well (Anna, Qatar)

The above indicates that participants in this research perceive experience to be a determinant in promotional success.

3.2.2. Cronyism

Cronyism, specifically a person's personal relationship with senior management in the school, was noted as a reason for promotion, as highlighted in the following quotes:

> Promotions are based on again your friendship circle, who you know and how you are seen with in the school. (Rose, Qatar)

> Sometimes degrees of nepotism or whatnot, you know who you know. (Patrick, Qatar)

> I think in this school a lot of it is to do with how well-liked people are by members of the senior leadership team (Aoibheann, Qatar)

These promotions were identified as specifically leading to perceptions amongst participants that leadership positions are being filled by individuals who are not capable of carrying out the role. This was exemplified in the following:

> Not everybody I know is being promoted that should have been promoted or some aren't particularly good at the job (Michelle, Qatar)

Not only did participants identify cronyism as a reason for being promoted, but at times, it was identified as a reason why new roles were created in the first place. Participants perceived that there were those who had been hand-selected for roles and that possible roles were sometimes written with a particular candidate in mind. One participant indicated strong perceptions in this regard:

> I think when opportunities come up at this school everyone already knows who's going to get it, it's kind of set-in stone (Aoibheann, Qatar)

> I think some people are kind of cherry picked to progress (Aoibheann, Qatar)

The perception that promotions were awarded due to cronyism was identified as having a negative impact on other employees. This is indicated in the following, where employees are discussed as being unhappy:

> Another peer member of staff from one of the other sites, he was promoted to senior leadership . . . some guys that were on the same level as him PE teacher you know doing that day-to-day teaching job, he got promoted and they were extremely unhappy because it was through connections that he had built, and he was approached for the job . . . but as a team of teachers you would know that very well, he was not deserving of that role but was approached and was basically given it (Emma, UK)

Furthermore, participants indicated that this may be detrimental to other employees whose value may not be appreciated, as highlighted in the following:

> Where you know there's a bit of a clique in school sometimes and there's certain people maybe those who allow those to get into positions of power and you see that all the time like you don't always see value in someone that's maybe a background worker. (Anthony, UK)

Finally, some participants perceived that some colleagues' promotions led to a detrimental effect on their career or their position in the school. One participant felt that they and others had to leave the school to escape the environment in which they were now working.

> A member of staff had also been asked if she would do the role before so she kind of always knew that she was second choice as well and I was in my NQT year so she just decided that she was going to act with any authority that she could

... She was left to just do whatever she wanted in her eyes and thought that she ran the whole of the infant school ... I left at the end of that school year and nine other members of staff left because of that one person well. (Aoibheann, Ireland)

The above findings highlight, from participants' perspectives, that promotions based on cronyism are taking place, and that this has detrimental consequences for employees who miss out on promotions and on employees' personal wellbeing due to subsequent unhappiness with promotion decisions. Additionally, promotions based on cronyism are perceived by participants to be resulting in school leaders who are unqualified for the position.

### 3.2.3. Gender

The gendered nature of promotions emerged repeatedly during interviews and was indicated by both male and female participants. A divide in school level emerged in this regard, with primary teachers highlighting that the majority of their middle leaders and senior management were female. Reasons given for this is that traditionally primary schools have had a more female-dominated workforce.

I would say traditionally a primary school well stereotypically a primary school teacher would be female years ago it's possibly seen as one of those gender stereotypical roles to be a female especially with younger children especially if you look down lower down the school towards FS [foundation stage] and key stage one with 100% makeup of this staff all being female. (Matthew, UK)

I think obviously in the primary school it's female dominated so I think that the numbers are there that eventually females will rise through the ranks (Allister, UK)

I would say in primary there generally tends to be more female teachers in primary and therefore there's a higher representation of females at a leadership level in primary. (Anna, Qatar)

Although the higher percentage of female leaders in primary roles was attributed to the higher number of female teachers at the primary level, at the secondary level, it was acknowledged that male teachers occupied a disproportionate number of leadership positions when considering the higher number of females in teaching roles at the same level:

I don't know if it exists everywhere, but I have seen I have seen it especially with SLT [Senior Leadership Team] that have been very male (dominated) and I'm not talking about where I am now, talking about another school that I worked at, not my last one, that was like very boy's club (Michelle, Qatar)

I think because there are so many less male primary teachers they get picked out of a pile, think it's like a peacock phenomenon (Aoibheann, Qatar)

On probing participants further during interview, when asked to cite reasons why a gender gap might exist, reasons given indicated societal norms and gender stereotypes such as the woman's role as primary caregiver in the home:

Some schools I think don't give people the jobs simply because they're females and from my experience females with children, they're going to see you as she's going to need time off for this and she's going to need time off for that so why promote a female and especially with kids when we've got a young guy who will bend over backwards to do what we want (Rose, Qatar)

Additional considerations for traditional stereotypes in terms of men being seen as more managerial or powerful than females were also highlighted, as can be seen in the following quotes:

It's just ingrained in you that if you see women in really powerful positions, it is outside of the norm (Anthony, UK)

It's the way of the world, isn't it? Men just have the opportunities and command the respect that women aren't necessarily always given, and I think that's a that's a problem that exists. (Kate, Qatar)

In the above, the participants of this study indicated that a gender division in promotions at the secondary school level appears to exist and favours male teachers. Reasons for this as identified by the participants were due to perceived traditional stereotypes such as women as carers and males as powerful and respected.

### 3.2.4. Credentials and Personality

Participants spoke about the need for credentials in terms of being able to avail themselves of promotional opportunities within a school context. Some participants viewed credentials, such as master's degrees, as the most important aspect of securing a promotion in a school:

> I've seen a lot of people who've gone away and done a master's programme or done a postgraduate program and they would always say that it's a benefit when you're going into an interview to be able to speak about that. (Barry, Ireland)

> Internal promotions, I've seen people get promoted because they have maybe achieved a particular qualification so that's allowed them to maybe go further. (Megan, Qatar)

> Principal(s) or the senior leadership need to have a master's (degree) in order to get that (role) (Patrick, Qatar)

Despite the importance of credentials, participants identified that one's personality can often hold equal, if not greater, sway in terms of who gets promoted. This was identified by multiple participants and exemplified in the following quotes:

> The domain factor is around an interview so personality and kind of how you how you put yourself across it's definitely going to be a factor and whether you seem like you're capable of doing it I suppose (Barry, Ireland)

> Sometimes if you have the credentials but I don't think if you don't have that ability to build relationships, you don't have that personality that kind of depending on what the role is that you're able to have those conversations maybe those difficult conversations. (Kate, Qatar)

> In terms of promotion obviously showing certain skills, that you have credentials to be a leader, personality again is a big thing (Elliot, Qatar)

The greater influence of personality over credentials was identified as misaligning with their own views on how promotions should be awarded. This was highlighted in the following quotes:

> I think maybe experience would be my number one and then I guess professionally it would probably be credentials and experience first and personality should be last, but I don't personally think that's how it's done all the time. (Elliot, Qatar)

The above from participants indicates that credentials are important for promotions, but personality is perceived at times to be a greater determinant for promotional success.

### 3.3. Work Relationship/Dynamics Stemming from Promotional Decisions

Work/relationship dynamics were a pertinent theme throughout. Most of the participants identified a shift in their work/relationship dynamics either at present or in the past, based on them being promoted or a colleague being promoted.

Difficulties arose in some instances for the middle leaders who were promoted within their own context. Participants indicated towards feelings of resistance, resentment, or lack of respect from colleagues:

> Just people not respecting your decision making process or questioning it or just when the message comes from someone else accepting it more than when it comes from you and I found it was a mixture of the fact that I started there as a PGCE (teacher placement) student and then also because I sometimes find it was

because I was a woman, because if a man said it, it was a little bit more palatable to them. (Kate, Qatar)

At one stage you know people maybe weren't taking me seriously as I wanted, like I had said, I had set a date for books to be in, they weren't in that day there were maybe two or three days late and then you know I'm expected to have them back a day later. (Allister, UK)

Since then, one teacher hardly looks at me, of course I make a point of all this you know saying hello and smiling at them, but I just know that they hate me like so that's the thing. (Patrick, Qatar)

Other participants spoke of working hard to maintain relationships once they have been promoted whilst recognising the difficulties that are presented in work/relationship dynamics once promoted:

It definitely changes, so like the dynamic of that group is going to change because you are the person that you've gone from being with them, essentially being one of them in that year group or whatever it is, to then being the leader, being the person that takes charge, makes decisions (Sarah, Qatar)

I've come across people, I've come across relationships in my previous school where two people went for a deputy principal job and one person like that was kind of was almost the shoo-in and like that his best friend had said look I'm going to go for the interview but even if I get it I'm not going to take it and subsequently he got it and ended up taking it so people like that who had been best friends on a personal level had kind of completely just become just a civil working relationship. (Barry, Ireland)

The unpleasant nature of work relationship/dynamics was also reiterated by two other participants, who felt a sense of resentment followed them or their colleagues after their promotion:

You work within a department, and you develop relationships and then you were suddenly above other people in terms of your position and department and for me it created lots of unhappy nights and lots of not so pleasant emails. (Rose, Qatar)

One individual was not gifted a promotion obviously earned the promotion, but the promotion wasn't advertised in terms of the same as every other phase within the school which has led to not so much negativity directly between the two but resentment. (Matthew, UK)

The above highlights that participants in this study perceive a potential for promotions to negatively impact on workplace relationships and negatively impact on school leaders' ability to do their work.

## 4. Discussion

The purpose of this study was to examine the perceptions of teachers and middle leaders in relation to promotional practices in schools. The results from this study highlighted common themes across teachers' and middle leaders' perceptions in relation to why people seek out promotions, why people do or do not receive promotions, and the potential dark side of promotions.

### 4.1. Why People Seek Out Promotions

Pay was the number one reason identified by the participants as to why teachers seek out promotional opportunities within schools. Interestingly, participants indicated that pay is a motivating factor to work harder for promotions and progression in the workplace. This reflects work by Rahim and Daud [55], who found that there was a relationship between extrinsic rewards and motivation to work harder. A rewards system, such as financial gain, could essentially motivate employees to maximise their work efforts. In this way, if

organisations implement and maintain adequate pay policies, they could attract high-quality candidates and retain their most valued and skilled staff members [55]. Ironically, however, it also emerged that despite the motivating factor of a pay rise, the actual pay was not always seen as comparable to the amount of work that was required by the position. This seemingly contradictory finding is one that would need to be addressed further in future studies but could be a result of pay being only one of two factors identified in this research.

Participants also alluded to ambition and a desire to be seen as more than just a teacher and to progress up the ladder to more senior positions. Previous research suggests that ambition can have a reasonable effect on a vast array of work and career behaviours and outcomes [56] such as job performance [57]. Ambition typically leads to people pursuing their goals in relation to accomplishment and striving for status within an organisation [58]. This aspect of ambition regarding promotion indicates an intrinsic motivation beyond that of financial gain. Furthermore, the motivation to rise higher in the ranks of the academic organisation suggests that promotion is a product of organisational culture in which organisational members seek out hierarchical progression [1]. Goals such as these require individuals to demonstrate a high-performance work ethic as a means to accomplish their goals in the hope of eventually satisfying their ambitions [59]. Recorded a total of 32 times within the results, ambition was a theme mentioned frequently by the respondents; moreover, the range of ambition-related responses was interesting, especially that of the aforementioned status of being "more than a teacher". This resonates with Hirschi and Spurk's [58] study wherein they state that ambition is positively associated with performance-related outcomes due to the various performance related behaviours that ambitious people undertake in the pursuit of their ambitious accomplishment and status goals. It is also interesting to note that motivations for school improvement or having a positive impact on the students or colleagues did not emerge as a motivation for seeking promotion. This is relevant as school leadership and student wellbeing are frequently discussed together [60–62] but did not emerge as a motivation for seeking out promotion. However, it is acknowledged that this could be as a result of this question not being specifically asked, and future studies may warrant further exploration.

### 4.2. Why People Do or Do Not Receive Promotions

In considering the findings pertaining to why people are promoted in schools, there emerged both perceivably fair and unfair reasons. In the interest of fairness, participants identified experience and credentials as factors influencing promotion decisions. This supports previous research indicating that experience is key to determining who is promoted within a workplace [63]. Despite this, there was significant agreement across participants to indicate perceptions of unfairness in promotional decisions. Reasons often cited within the findings of this study for people being promoted were friendship promotions. Other terms for this are favouritism, nepotism, and cronyism and have emerged in other studies as main barriers to employment and promotions of well-qualified employees [64]. Favouritism is defined as the recruitment, appraisal, and promotion on the basis of whom you know rather than your experience, qualifications, and merit [65]. Stahl et al. [66] speak of the professionalization of organisations and the expectation that managers are recruiting on merit, qualifications, and experience. However, as Oyserman and Markus [67] point out, the problem with smaller and more communal organisations, such as schools in the case of this study, remains stuck within a system whereby obligations to workers feature. These obligations and expectations often exert pressure on hiring leaders to diverge from professional standards and instead resort to cronyism, nepotism, and favouritism for an easier life [68]. In this study, a sense of unjustness was identified amongst teachers regarding promotional and hiring practices within their current context or indeed in previous contexts, echoing findings from Shabbir and Siddique [64]. Most of the teachers had experienced some form of favouritism when it came to hiring policies within their school. In support of Prendergast and Topel [65], at times there was a sense amongst teachers that jobs or roles were created with particular people in mind. Of particular interest in the findings was the

identification of instances in which the job specification would be written for a particular person or there would be certain specification points that would discourage certain people from applying in favour of others. There is strong evidence from these findings, therefore, to support the existence of favouritism, or at least a perception of favouritism in promotion processes in schools. This in turn can have influence whether teachers will even apply for roles going forward.

An additional area of concern identified in this research was the sense that male teachers may be more likely to be promoted than female ones. Although this was not acknowledged at the primary level, it was prevalent in secondary level staff. It was noted even more so at the secondary level that more males than females were in roles at the senior leadership level (principal and vice principal), and that often it could be quite difficult for females to make the leap to senior leadership level in a secondary school. This is supported by the literature relating to a study carried out in the United Kingdom by the Department for Education [69], wherein females made up 85% of the workforce in primary schools and 73% of the leadership roles in comparison to secondary schools, wherein females made up 62% of the workforce but only 38% of leadership roles. One reason for this which emerged from the findings was gendered societal roles: it is ingrained within us to not typically see women in positions of authority or high levels of management. This agrees with numerous research findings in a wide range of fields, particularly on the topic of the "think manager, think male" stereotype [70]. This finding is particularly interesting as it represents traditional gender stereotypes prevailing in schools and influencing opportunities for promotion. This is supported by Ginige, Amaratunga, and Haigh [71], who suggest that societal roles and gender equality, while more recognised now than previously, still exist in our modern society and culture, and women are still more negatively impacted than males. Even more than males being seen more as manager material, it also emerged that the caring role often aligned with females may also exist and further prohibit women from promotional opportunities. This was outlined by the participants of the study who alluded to the fact that females are not promoted as they will need time off to look after sick children, which again supports previous literature pertaining to the socialization of women into more caring roles [72]. Furthermore, it supports research that has identified the caring responsibilities of women as negatively impacting their promotional opportunities [73]. These findings go some way toward explaining why, despite dominating the primary and secondary teaching profession across the globe, women remain underrepresented in leadership and management roles in both public and private schools at secondary level [74].

*4.3. The Dark Side of School Promotions*

It was highlighted within these findings that work relationships/dynamics and the difficulties that were presented by these decisions was evident, both by those who have been promoted and those who have had former peers promoted to a level above them. Participants spoke of peers who felt that they had to throw their weight around or rule with an iron fist in order to be respected, whilst others who were promoted spoke about the upsetting nature of dealing with staff who were once previously at the same level as them whom they were now managing. Others felt that positive relationships with their managers or subordinates was possible following promotional decisions; however, not one example of this could be provided. This was completely down to personal experience, personal relationships, and indeed personality of the managers and subordinates. The literature states that subordinates are much more amenable to accepting positively framed feedback from supervisors than negative feedback [75,76]. This suggests that the nature of the feedback would impact the relationship between manager and subordinate and focus solely on how it is being delivered, as opposed to what is being delivered. Recognising that negative feedback is sometimes a prerequisite of a promotional role, Trope and Neter [77] state that it is more valuable to an organisation than constant positive feedback. While this was recognised by some of the participants, they felt that feedback, even if negative, needed to be framed in a constructive and professional manner.

The breakdown in the relationship between co-workers following promotion as indicated in the findings led to some of the participants being particularly upset. Sleep disturbance, receiving emails out of working hours (work–life spill-over), and relationship difficulties were reported by the participants. Alarmingly, turnover intentions and actual turnover were identified because of poor workplace promotions. Harvey et al. [78] have stated that a leader's ability to interact successfully and maintain high-quality relationships with their subordinates is crucial to ensuring the success of an organisation. The findings identified turnover of staff as a possible negative outcome of poor promotional decisions due to personality conflicts within a department or year group. Looking at LMX theory, the quality of the relationships of the leader–members can range from low, wherein the relationship is exclusively focused on employment, to high-quality relationships where social exchange, liking, trust and respect are present [79]. The research, although limited in scope, indicated that instances where low-quality LMX was present resulted in participants vacating their roles due to the difficulties experienced with either their subordinates or their managers.

## 5. Conclusions and Recommendations

The findings of this study indicate that there is at least a perception of unfairness in promotional practices in schools at primary and secondary levels. As indicated by the findings and supported by previous research, this can have a negative impact on the wellbeing of both the promoted and the unpromoted. To mitigate this, schools may benefit from ensuring that the focus on work promotions is not only focused predominantly on perceivably fair factors such as experience and credentials, but that this focus is clearly communicated to staff. Additionally, perceptions of unfair processes must be tackled. If it is the case that these perceptions are true, then greater processes for ensuring fairness in promotions should be implemented. If unfounded, then it is of importance to also highlight that they are not persistent. These measures may reduce the perception of unfairness and promote a more sustainable culture and greater staff wellbeing. Additionally, there is an indication in the findings towards a gender bias in promotional practices that schools need to ensure is being addressed and not prevalent. This is in the interest of equality, inclusion, and tackling the gender gap in promotions.

Despite the important findings that have emerged from this study, it is acknowledged that it is based on perceptions and experiences of a perceivably small cohort. This study from 15 participants across different schools and countries may set the foundations for future research to further explore the findings which emerged in this scoping, exploratory study. The qualitative research design allowed for in-depth exploration of perceptions and experiences of the participants; however, as with most qualitative studies, the risk of generalising the wider population remains high. Additionally, the current research did not find evidence of demographical differences between participants' responses; however, this could be as a result of the sample size. For that reason, future researchers may consider the adoption of a quantitative approach through use of questionnaires to explore the accuracy of the findings on a broader scale. This would also allow for the consideration of a variety of factors including, but not limited to, gender, age differentials, culture differences, school size, and career stage, which may also play a role.

Furthermore, as this study alluded to the perceptions of unfairness and gender biases in school promotions, it is important that these factors are explored in greater detail. This is not only of importance and responsibility to the academic community, but to the wider education systems in which primary and post-primary schools are subsumed.

**Author Contributions:** Conceptualization, R.H. and N.L.; methodology, R.H. and N.L.; formal analysis, R.H.; investigation, R.H.; writing—original draft preparation, R.H., N.L. and P.M.M.; writing—review and editing, R.H., N.L. and P.M.M.; visualization, N.L.; supervision, N.L.; project administration, N.L. All authors have read and agreed to the published version of the manuscript.

**Funding:** This research received no external funding.

**Institutional Review Board Statement:** The research discussed in this paper received ethical approval from the Education and Health Sciences Ethics Department, University of Limerick. Approval number: EHSREC no 2021_12_11_EHS.

**Informed Consent Statement:** Informed consent was obtained from all participants in this study.

**Data Availability Statement:** Data available on request due to restrictions placed on participant privacy, however, segments of the data can be requested by contacting the corresponding author.

**Conflicts of Interest:** The authors declare no conflict of interest.

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
