# Peer review of "Leadership Opportunities in the School Setting: A Scoping Study on Staff Perceptions"

_societies, doi:10.3390/soc13050129_

Round 1
Reviewer 1 Report
Please see the attachment

Author Response
Thank you very much for your comments. The team and I thoroughly welcomed your very informative and constructive feedback. We hope that we have adequately addressed all of your comments. The changes we have made (and responses to some of you questions) are highlighted in the attached document.

Reviewer 2 Report
The evaluated manuscript deals with the perspective of promotion of educational leaders and various organizational practices in the school. It is a subject that can be interesting from the point of view of the educational organization. However, in the introduction to reality and in the literature review carried out, educational leadership or the promotion of workers in this area are not addressed in any way. It delves into various aspects necessary for research, but with the perspective of other areas of knowledge, not that of the school organization.
On the other hand, deficiencies are also detected in the methodological aspects of the manuscript. Essential aspects such as research design, research quality criteria or ethical research issues are not established. In addition, the results presented are scarcely analyzed, only an excessive number of citations categorized by the researchers are shown. This part should be developed more.
Finally, an adequate discussion of the research carried out is proposed. However, the conclusions are made vaguely and without any interest.
Author Response
The team and I would like to thank you for your informative and constructive feedback. We hope that we have addressed all of your concerns and queries in the attached document.

Round 2
Reviewer 2 Report
I would like to congratulate the authors for the improvements made to the manuscript. The set of research presented has undergone considerable improvements. However, I would like to stress again the presentation of the results, which must be modified. Although the analysis process is described, qualitative research requires more depth in the interpretation of the data, not just listing the citations. I understand what you are saying in your answer when you point out that the citations are given to support the findings. However, the findings are not so many because of the problem that I am indicating. It seems appropriate to me that phase 6 of Creswell's method of analysis should be worked on more deeply, since it is carried out in a very vague way.
Author Response
Dear Reviewer,
The team would like to again thank you for taking the time to review our paper and for your response. On revisiting the results section we completely agree with your comments and have made major revisions to this section. The changes show (we believe) greater engagement and interpretation of the data. Concluding sentences have been added to the end of each sub section also as a means to summarize the reported findings within that section.
I have attached the newly structured results section as a word file.
Kind regards,
Niamh

Round 3
Reviewer 2 Report
It is noticeable that changes have been made in the results section. However, these changes do not improve the shortcomings observed in the results.
Author Response
The demographic table in the methodology section has been updated to include gender, pseudonyms, school position, where participants studied, country they work, and school type. The number of participants who referred to each theme/finding are now included.
The methodology chapter outlines how the data was analysed and how findings/themes emerged.
The updated results section presents the findings, along with the additional information as highlighted in the above paragraph.
The discussion section presents an interpretation of the results in consideration of the existent literature in the field.